# Exploiting River Network Topology for Flood Forecasting with Graph Neural Networks

## Abstract

Climate change exacerbates riverine floods, which occur with higher frequency and intensity than ever. The much-needed forecasting systems typically rely on accurate river discharge predictions. To this end, the SOTA data-driven approaches treat forecasting at spatially distributed gauge stations as isolated problems, even within the same river network. However, incorporating the known river network topology into the prediction model has the potential to *leverage the adjacency relationship between gauges*. Thus, we model river discharge for a network of gauging stations with a GNN, and compare the forecasting performance achieved by different adjacency definitions. Our results show that the model fails to benefit from the river network topology information, regardless of the number of layers and, thus, propagation distance. The learned edge weights correlate with neither of the static definitions and exhibit no regular pattern. Furthermore, a worst-case analysis reveals that the GNN struggles to predict sudden discharge spikes. This work may serve as a justification for the SOTA treating gauges independently and suggests that more improvement potential lies in anticipating spikes.

## 1 Introduction

Floods are among the most destructive natural disasters that occur on Earth, causing extensive damage to infrastructure, property, and human life. They are also the most common type of disaster, accounting for almost half of all disaster events recorded (cp. Figure 1). In 2022 alone, floods affected 57.1 million people worldwide, killed almost 8000, and caused 44.9 billion USD in damages (CRED, 2022). With climate change ongoing, floods have become increasingly frequent over the last decades and are expected to be even more prevalent in the future (United Nations, 2022). Thus, early warning systems that can help authorities and individuals prepare for and respond to impending floods play a crucial role in mitigating fatalities and economic costs.

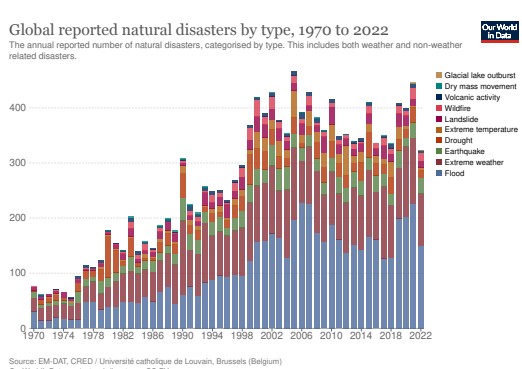

Figure 1: Historical occurrence of natural disasters by disaster type. The number of events increased over time, with floods being the most common. (Ritchie et al., 2022).

Operational forecasting systems such as Google's Flood Forecasting Initiative (Nevo et al., 2022) typically focus on riverine floods, which are responsible for the vast majority of damages. A key component in these systems is the prediction of future river discharge[1] at a gauging station based on environmental indicators such as past discharge and precipitation. The state-of-the-art data-driven approaches are based on Kratzert et al. (2019b) and consist in training an LSTM variant on multiple gauges jointly to exploit the shared underlying physics. However, even when some of those gauges are in the same river network, this topology information is not taken into account. One reason might be that the main benchmarking dataset family CAMELS-x (Addor et al., 2017; Alvarez-Garreton et al., 2018; Coxon et al., 2020; Chagas et al., 2020; Fowler et al., 2021) does not contain such information. Recently, Klingler et al. (2021) published a new benchmarking dataset LamaH-CE that follows the CAMELS-x framework but includes topology data.

In this work, we investigate the effect of river network topology information on discharge predictions by employing a single end-to-end GNN to allow the network structure to be utilized during the prediction process. We train GNNs on LamaH-CE and, to assess the merit of incorporating the graph structure, compare the effect of different adjacency definitions:

(1) no adjacency, which is equivalent to existing approaches with cross-gauge shared parameters but isolated gauges,

(2) binary adjacency of neighboring gauges in the network,

(3) weighted adjacency according to physical relationships like stream length, elevation difference, and average slope between neighboring gauges, and

(4) learned adjacency by treating edge weights as a model parameter.

Furthermore, we inspect how the learned edge weights from (4) correlate with the static weights in (3). We also explore the role of information propagation distance on predictive capabilities and analyze the model's behavior on the worst-performing gauge. Our source code is publicly available at `https://add-link-after-review`.

## 2 RELATED WORK

Classical approaches towards river discharge prediction stem from finite-element solutions to partial differential equations such as the Saint-Venant shallow-water equations (Vreugdenhil, 1994; Wu, 2007). However, these models suffer from scalability issues since they become computationally prohibitive on larger scales, as required in the real world (Nevo et al., 2020). Furthermore, they impose a strong inductive bias by making numerous assumptions about the underlying physics.

On the other hand, data-driven methods and in particular deep learning provide excellent scaling properties and are less inductively biased. They are increasingly being explored for a plethora of hydrological applications, including discharge prediction (see surveys by Mosavi et al., 2018; Chang et al., 2019; Sit et al., 2020), where they tend to achieve higher accuracy than the classical models. The vast majority of studies employ Long Short-Term Memory models (LSTM; Hochreiter & Schmidhuber, 1997) due to their inherent suitability for sequential tasks and reliability in predicting extreme events (Frame et al., 2022). Whereas these studies usually consider forecasting for a single gauging station, Kratzert et al. (2019a;b) demonstrate the generalization benefit of training a single spatially distributed LSTM model on multiple gauging sites jointly. Their approach exploits the shared underlying physics across gauges but is still agnostic to the relationship between sites.

Incorporating information from neighboring stations or even an entire river network into a spatially distributed model may improve prediction performance. Upstream gauges could "announce" the advent of significantly increased water masses to downstream gauges, which in turn could provide forewarning about flooding already ongoing further downstream. The input then becomes a graph whose vertices represent gauges and edges represent flow between gauges. The corresponding deep learning tool to capture these spatial dependencies is Graph Neural Networks (GNN). Kratzert et al. (2021) employ such a GNN as a post-processing step to route the per-gauge discharge predicted by a conventional LSTM along the river network, but it does not perform the actual prediction.

---

[1] amount of water volume passing through a given river section per unit time

## 3  METHODOLOGY

### 3.1  DATA PREPROCESSING

The LamaH-CE[2] dataset (Klingler et al., 2021) contains historical discharge and meteorological measurements on an hourly resolution for 859 gauges in the broader Danube river network shown in Figure 2. Covering an area of $170\,000\,\text{km}^2$ with diverse environmental conditions, Klingler et al. expect that results from investigations on this dataset carry over to other river networks. One caveat is that LamaH-CE does not provide any flood event annotations, so that we can only model continuous discharge but not floods as discrete events.

The river network defined by LamaH-CE naturally forms a directed acyclic graph (DAG) $\mathcal{G} = (\mathcal{V}, \mathcal{E})$. The nodes $\mathcal{V}$ represent gauges, and the edges $\mathcal{E}$ represent flow between a gauge and the next downstream gauges. Hence, $\mathcal{G}$ is *anti-transitive*, i.e., no skip connections exist. We preprocess $\mathcal{G}$ to distill a connected subgraph with complete data.

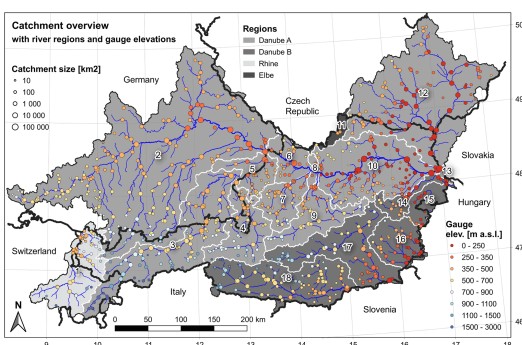

Figure 2: Geographical overview of LamaH-CE. Circle color indicates gauge elevation; circle size indicates catchment size. (Klingler et al., 2021)

**Region Selection.** Figure 2 shows that $\mathcal{G}$ contains four different connected components, of which we restrict ourselves to the largest one, "Danube A". Its most downstream gauge close to the Austrian-Hungarian border has complete discharge data for the years 2000 through 2017. Starting at this gauge, we determine all connected gauges of the Danube A region by performing an inverse depth-first search given by Algorithm A.1. Overall, 608 out of the original 859 gauges belong to this connected component.

**Gauge Filtering.** While the meteorological data is complete, the discharge data contains gaps. Klingler et al. have filled any consecutive gaps of at most six hours by linear interpolation and left the remaining longer gaps unaltered. We only want to consider gauges that (a) do not have these longer periods of missing values and (b) provide discharge data for at least the same time frame (2000 to 2017) as the most downstream gauge. To this end, we remove all gauges that violate these requirements from the graph using Algorithm A.2. Predecessors and successors of a deleted node get newly connected so that network connectivity is maintained. Note that thanks to antitransitivity, a duplicate check is unnecessary when inserting the new edges. After this preprocessing step, we are left with 375 out of the previously 608 gauges.

Overall, the reduced graph $\mathcal{G}$ now consists of $n := |\mathcal{V}| = 375$ gauges with $T$ hours of discharge measurements for the years 2000 to 2017, which we can conceptually represent as a node signal $\mathbf{Q} = \left[\boldsymbol{q}^{(1)} \mid \boldsymbol{q}^{(2)} \mid \ldots \mid \boldsymbol{q}^{(T)}\right] \in \mathbb{R}^{n \times T}$. This cleaned dataset needs to be prepared for training.

**Normalization.** As is common practice in deep learning, we normalize the data to surrender all gauges to the same scale and accelerate the training process (LeCun et al., 2002). In particular, we normalize per gauge (i.e., element-wise) using the standard score:

$$\boldsymbol{\mu} = \frac{1}{T}\sum_{t=1}^{T}\boldsymbol{q}^{(t)}, \qquad \boldsymbol{\sigma}^2 = \frac{1}{T-1}\sum_{i=1}^{T}(\boldsymbol{q}^{(t)} - \boldsymbol{\mu})^2, \qquad \boldsymbol{q}^{(t)} \leftarrow \frac{\boldsymbol{q}^{(t)} - \boldsymbol{\mu}}{\boldsymbol{\sigma}}$$

**Train-test splits.** To robustly assess the performance of a trained model on unseen data via cross-validation, we randomly partition the 18 available years of observations into six folds of three years. By choosing one fold as the test set and the remaining folds as the training set, we obtain six different train-test splits that we keep constant throughout experiments.

---

[2]**LA**rge-Sa**M**ple **DA**ta for **H**ydrology for **C**entral **E**urope

## 3.2 THE FORECASTING TASK

We task the model with an instance of supervised node regression. Assume we are given a certain amount of $W$ (*"window size"*) most recent hours of discharge and meteorological measurements, in particular precipitation, topsoil moisture, air temperature, and surface pressure, for all gauges. Our goal is to predict the discharge $L$ (*"lead time"*) hours in the future. For simplicity, we restrict the following illustrations to the discharge data in the input since the meteorological data can be trivially added in an extra dimension.

**Features & Targets.** To conduct supervised learning, we extract input-output pairs from the time series represented by $\mathbf{Q}$ (cp. Section 3.1). For $t = W, W + 1 \dots, T - L$, we define the feature matrix at time step $t$ and the corresponding target vector as

$$\mathbf{X}^{(t)} := \left[ \boldsymbol{q}^{(t-W+1)} \;\middle|\; \dots \;\middle|\; \boldsymbol{q}^{(t-1)} \;\middle|\; \boldsymbol{q}^{(t)} \right] \in \mathbb{R}^{n \times W}, \qquad \boldsymbol{y}^{(t)} := \boldsymbol{q}^{(t+L)} \in \mathbb{R}^n.$$

We collect all samples into the set $\mathcal{D} = \{(\mathbf{X}^{(t)}, \boldsymbol{y}^{(t)})\}_{t=W}^{T-L}$ and partition it according to a given train-test split into $\mathcal{D} = \mathcal{D}_{\text{train}} \uplus \mathcal{D}_{\text{test}}$. The extraction process can be illustrated as follows:

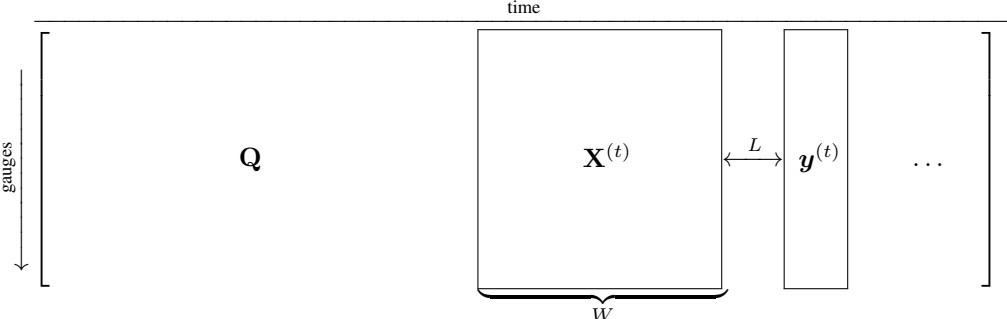

**Adjacency.** Besides the input and target measurements, we feed the river network topology to the GNN in the form of an adjacency matrix $\mathbf{A} \in \mathbb{R}^{n \times n}$. For the definition of matrix entries corresponding to an edge $(i, j) \in \mathcal{E}$ (the rest being zero), we consider the following choices:

(1) *isolated:* $\mathbf{A}_{i,j} := 0$ equates to removing all edges and results in the augmented normalized adjacency matrix to be a multiple of the identity so that each GNN layer degenerates to a node-wise linear layer.

(2) *binary:* $\mathbf{A}_{i,j} := 1$ corresponds to the unaltered adjacency matrix as it comes with the LamaH-CE dataset.

(3) *weighted:* $\mathbf{A}_{i,j} := w_{(i,j)}$ quantifies a physical relationship, for which LamaH-CE provides three alternatives:

- the *stream length* along the river between $i$ and $j$,
- the *elevation difference* along the river between $i$ and $j$, and
- the *average slope* of the river between $i$ and $j$.

(4) *learned:* $\mathbf{A}_{i,j} := \omega_{(i,j)}$ where $\boldsymbol{\omega} \in \mathbb{R}^{|\mathcal{E}|}$ is a learnable model parameter.

The first two variants allow us to compare the effect of introducing the river network topology into the model at all. The last two variants enable insights into what kind of relative importance of edges is most helpful. As usual in GNNs, we define the normalized augmented adjacency matrix

$$\bar{\mathbf{A}} := (\mathbf{D}_{\text{in}} + \text{diag}(\boldsymbol{\lambda}))^{-\frac{1}{2}} (\mathbf{A} + \text{diag}(\boldsymbol{\lambda})) (\mathbf{D}_{\text{in}} + \text{diag}(\boldsymbol{\lambda}))^{-\frac{1}{2}}$$

where self-loops for node $i$ with weight $\lambda_i$ are added and everything is symmetrically normalized based on the diagonal in-degree matrix $\mathbf{D}_{\text{in}}$. We generally set $\lambda_i$ as the mean of all incoming edge weights at node $i$ to make self-loops roughly equally important to the other edges. The only exception to this is option (1) above, where that mean would be zero and thus result in no information flow whatsoever, so that in this case, we set the self-loop weights to one instead.

**Model.** Our desideratum is a GNN $f_\theta : \mathbb{R}^{n \times W} \to \mathbb{R}^n$ parameterized by $\theta$ which closely approximates the mapping of windows $\mathbf{X}$ to targets $\boldsymbol{y}$, i.e., $\hat{\boldsymbol{y}} \coloneqq f_\theta(\mathbf{X}) \approx \boldsymbol{y}$. All our models have a sandwich architecture: a linear layer $\mathrm{Encoder}_{\boldsymbol{\Theta}_0} : \mathbb{R}^{n \times W} \to \mathbb{R}^{n \times d}$ embeds the $W$-dimensional input per gauge into a $d$-dimensional latent space. On this space, a sequence of $N$ layers $\mathrm{GNNLayer}_{\boldsymbol{\Theta}_i} : \mathbb{R}^{n \times d} \times \mathbb{R}^{n \times n} \to \mathbb{R}^{n \times d}$ are applied. Finally, another linear layer $\mathrm{Decoder}_{\boldsymbol{\Theta}_{N+1}} : \mathbb{R}^{n \times d} \to \mathbb{R}^n$ projects from the latent space to a scalar per gauge. In symbols:

$$\mathbf{H}^{(0)} \coloneqq \mathrm{Encoder}_{\boldsymbol{\Theta}_0}(\mathbf{X})$$
$$\mathbf{H}^{(i)} \coloneqq \mathrm{GNNLayer}_{\boldsymbol{\Theta}_i}(\mathbf{H}^{(i-1)}, \bar{\mathbf{A}}) \quad \text{for } i = 1, \dots, N$$
$$\hat{\boldsymbol{y}} \coloneqq \mathrm{Decoder}_{\boldsymbol{\Theta}_{N+1}}(\mathbf{H}^{(N)}).$$

We consider three choices for GNNLayer, with $\sigma = \mathrm{ReLU}$ as activation function:

$$\mathrm{GCNLayer}_{\boldsymbol{\Theta}}(\mathbf{H}, \bar{\mathbf{A}}) \coloneqq \sigma(\bar{\mathbf{A}}^\top \mathbf{H}\boldsymbol{\Theta}) \qquad \text{(Kipf \& Welling 2017)}$$
$$\mathrm{ResGCNLayer}_{\boldsymbol{\Theta}}(\mathbf{H}, \bar{\mathbf{A}}) \coloneqq \mathbf{H} + \mathrm{GCNLayer}_{\boldsymbol{\Theta}}(\mathbf{H}, \bar{\mathbf{A}})$$
$$\mathrm{GCNIILayer}_{\boldsymbol{\Theta}}(\mathbf{H}, \bar{\mathbf{A}}) \coloneqq \sigma\big(\big((1-\alpha)\bar{\mathbf{A}}^\top \mathbf{H} + \alpha\mathbf{H}^{(0)}\big)\big((1-\beta)\mathbf{I} + \beta\boldsymbol{\Theta}\big)\big) \quad \text{(Chen et al. 2020)}$$
$$\text{where } \alpha, \beta \in (0, 1)$$

While the vanilla GCNLayer is the simplest definition, it famously suffers from a phenomenon known as *oversmoothing* (Oono & Suzuki, 2020) where the features of adjacent nodes converge with increasing depth. To alleviate this undesirable behavior, ResGCNLayer adds a residual connection, whereas GCNIILayer introduces the notions of initial connection and identity mapping via weighted averages.

**Optimization Objective.** To measure the error between a model prediction $\hat{\boldsymbol{y}}$ and the target $\boldsymbol{y}$, we use the multi-dimensional square loss $L(\hat{\boldsymbol{y}}, \boldsymbol{y}) \coloneqq \frac{1}{n}\|\hat{\boldsymbol{y}} - \boldsymbol{y}\|^2$. Training is then defined as optimizing the expected loss over the empirical distribution of training samples in $\mathcal{D}_{\text{train}}$, i.e., the optimal model parameters are given by

$$\arg\min_\theta \mathbb{E}_{(\mathbf{X}, \boldsymbol{y}) \sim \mathcal{D}_{\text{train}}}[L(f_\theta(\mathbf{X}, \bar{\mathbf{A}}), \boldsymbol{y})].$$

**Metrics.** Recall that we perform training on normalized samples. For evaluation, we must calculate metrics on the unnormalized version of the predictions and targets:

$$\hat{\boldsymbol{y}}_{\text{orig}} \coloneqq \boldsymbol{\sigma} \odot \hat{\boldsymbol{y}} + \boldsymbol{\mu}, \qquad \boldsymbol{y}_{\text{orig}} \coloneqq \boldsymbol{\sigma} \odot \boldsymbol{y} + \boldsymbol{\mu}.$$

The most intuitive regression metric is the *Mean Squared Error* (MSE). In our multi-dimensional regression problem, it is defined as the error vector

$$\mathbf{MSE} \coloneqq \frac{1}{|\mathcal{D}_{\text{test}}|} \sum_{i=1}^{|\mathcal{D}_{\text{test}}|} (\hat{\boldsymbol{y}}_{\text{orig}}^{(t_i)} - \boldsymbol{y}_{\text{orig}}^{(t_i)})^2 = \boldsymbol{\sigma}^2 \odot \frac{1}{|\mathcal{D}_{\text{test}}|} \sum_{i=1}^{|\mathcal{D}_{\text{test}}|} (\hat{\boldsymbol{y}}^{(t_i)} - \boldsymbol{y}^{(t_i)})^2.$$

Next to the MSE, a standard metric in hydrology is the *Nash-Sutcliffe Efficiency* (NSE; Nash & Sutcliffe, 1970). It compares the sum of squared errors of the model to the sum of squared errors of the constant mean-predictor and subtracts this value from one to obtain a percentage score in $[0, 1]$. An NSE of zero means that the model's predictive capability is no better than that of the empirical mean, while an NSE of one means that all model predictions are perfect.

$$\mathbf{NSE} \coloneqq 1 - \frac{\sum_{i=1}^{|\mathcal{D}_{\text{test}}|} (\hat{\boldsymbol{y}}_{\text{orig}}^{(t_i)} - \boldsymbol{y}_{\text{orig}}^{(t_i)})^2}{\sum_{i=1}^{|\mathcal{D}_{\text{test}}|} (\boldsymbol{\mu} - \boldsymbol{y}_{\text{orig}}^{(t_i)})^2} = 1 - \frac{\mathbf{MSE}}{\boldsymbol{\sigma}^2}$$

We straightforwardly obtain summary metrics for our experiments by averaging across gauges:

$$\overline{\mathrm{MSE}} \coloneqq \frac{1}{n} \sum_{g=1}^n \mathrm{MSE}_g, \qquad \overline{\mathrm{NSE}} \coloneqq \frac{1}{n} \sum_{g=1}^n \mathrm{NSE}_g.$$

## 4 EXPERIMENTS

### 4.1 EXPERIMENTAL SETUP

The code to reproduce our experiments is publicly available[3]. Table 1 lists the relevant hyperparameters we use throughout all experiments unless stated otherwise, categorized into data, model, and training parameters.

On the data side, we choose a window size of $W = 24$ hours as a compromise between sufficiently many past observations and computational efficiency. We set the lead time to $L = 6$ hours, which is a realistic choice.

On the model side, we consider all three choices of layer definition detailed in Section 3.2, resulting in three model architectures GCN, ResGCN, and GCNII. We choose a depth of $N = 20$ layers to allow information propagation along the entire river graph, given that the longest path in the preprocessed graph consists of 19 edges. The latent space dimensionality of $d = 128$ was chosen large enough to allow an injective feature embedding but small enough to avoid memory issues. The edge direction and adjacency type hyperparameters will be explored in detail in Section 4.2.

Table 1: Default hyperparameter choices for our experiments.

|  | HYPERPARAMETER | VALUE |
|---|---|---|
| DATA | WINDOW SIZE ($W$) | 24 h |
|  | LEAD TIME ($L$) | 6 h |
|  | NORMALIZATION? | YES |
| MODEL | ARCHITECTURE | [RES]GCN, GCNII |
|  | NETWORK DEPTH ($N$) | 20 |
|  | LATENT SPACE DIM ($d$) | 128 |
|  | EDGE DIRECTION | BIDIRECTIONAL |
|  | ADJACENCY TYPE | BINARY |
| TRAINING | INITIALIZATION | GLOROT |
|  | OPTIMIZER | ADAM |
|  | # EPOCHS | 20 |
|  | BATCH SIZE | 64 |
|  | LEARNING RATE | $5 \times 10^{-4}$ |

On the optimization side, all neural network parameters are randomly initialized using the standard Glorot initialization scheme (Glorot & Bengio, 2010). We then perform 20 epochs of stochastic mini-batch gradient descent, which is enough for the process to converge. The descent algorithm is Adaptive Moments (Adam) (Kingma & Ba, 2015) with a base learning rate of $5 \times 10^{-4}$, which results in stable training. To prevent overfitting, we randomly hold out 1/5 of the training set, which corresponds to three years of observations, and select the parameters from the epoch in which the loss calculated over this holdout set was the lowest.

### 4.2 RIVER TOPOLOGY COMPARISON

Our main experiment compares the impact of the six different gauge adjacency definitions detailed in Section 3.2 on forecasting performance. In addition, we also consider three alternative edge orientations, which determine the direction of information flow in the GNN, as none of the options is a priori preferable. The *downstream* orientation is given by the dataset, the *upstream* orientation results from reversing all edges, and the *bidirectional* orientation from adding all reverse edges to the forward ones. We six-fold cross-validate all 18 topology combinations using the train-test splits established in 3.1 and the average MSE and NSE metrics defined in Section 3.2 and report the results for ResGCN and GCNII in Table 2. As the vanilla GCN suffers heavily from oversmoothing, we disregard it in the remaining discussions and only provide its results in Table A.2 for completeness.

Surprisingly, model performance for ResGCN and GCNII shows almost no sensitivity to the choice of graph topology. Isolating the gauges does not harm performance beyond the standard deviation, and no combination outperforms a 20-layer MLP baseline by a margin. This indicates that the forecasting task for a gauge mainly benefits from the past discharge at that gauge but not from the discharge at neighboring gauges. The river graph topology makes no difference. Even when the model is allowed to learn an optimal edge weight assignment, it does not manage to outperform the baseline. However, a consistent pattern is that the GNNs achieve their best average NSE for a bidirectional edge orientation.

---

[3] https://add-link-after-review

Table 2: Forecasting performance on different river network topologies, given as mean and standard deviation of the respective metrics across folds. $\overline{\text{MSE}}$ is not scale-normalized per gauge, while $\overline{\text{NSE}}$ is (cp. Section 3.2). A 20-layer MLP baseline achieves an NSE of $85.62\% \pm 4.90\%$. Bold indicates the best value per column. Note that results for the isolated adjacency type are not affected by the choice of edge orientation due to the absence of edges in this case.

(a) ResGCN

| | DOWNSTREAM | | UPSTREAM | | BIDIRECTIONAL | |
|---|---|---|---|---|---|---|
| ADJACENCY TYPE | $\overline{\text{MSE}}\downarrow$ | $\overline{\text{NSE}}\uparrow$ | $\overline{\text{MSE}}\downarrow$ | $\overline{\text{NSE}}\uparrow$ | $\overline{\text{MSE}}\downarrow$ | $\overline{\text{NSE}}\uparrow$ |
| ISOLATED | 899.80 ±1329.17 | 80.85 % ±11.66 % | 899.80 ±1329.17 | 80.85 % ±11.66 % | 899.80 ±1329.17 | 80.85 % ±11.66 % |
| BINARY | 353.54 ±80.90 | **83.53 %** ±**5.63 %** | 372.67 ±61.11 | 84.99 % ±5.10 % | 741.20 ±166.26 | 85.34 % ±4.86 % |
| STREAM LENGTH | 524.03 ±100.46 | 83.42 % ±5.59 % | 435.66 ±60.49 | 84.74 % ±5.02 % | 785.38 ±171.49 | 85.31 % ±4.92 % |
| ELEVATION DIFFERENCE | 407.67 ±95.16 | 83.46 % ±5.60 % | 456.32 ±63.80 | 83.76 % ±4.80 % | 773.95 ±182.22 | 85.16 % ±4.93 % |
| AVERAGE SLOPE | **327.22** ±**75.81** | 83.45 % ±5.60 % | 425.95 ±86.43 | 84.10 % ±5.18 % | 656.52 ±170.12 | 85.23 % ±4.92 % |
| LEARNED | 345.57 ±199.76 | 83.50 % ±5.40 % | **366.94** ±**80.72** | **85.63 %** ±**4.65 %** | **567.39** ±**160.84** | **85.94 %** ±**4.52 %** |

(b) GCNII

| | DOWNSTREAM | | UPSTREAM | | BIDIRECTIONAL | |
|---|---|---|---|---|---|---|
| ADJACENCY TYPE | $\overline{\text{MSE}}\downarrow$ | $\overline{\text{NSE}}\uparrow$ | $\overline{\text{MSE}}\downarrow$ | $\overline{\text{NSE}}\uparrow$ | $\overline{\text{MSE}}\downarrow$ | $\overline{\text{NSE}}\uparrow$ |
| ISOLATED | 289.71 ±50.01 | 85.95 % ±4.97 % | 289.71 ±50.01 | 85.95 % ±4.97 % | **289.71** ±**50.01** | 85.95 % ±4.97 % |
| BINARY | 277.50 ±33.57 | **86.17 %** ±**4.69 %** | 312.31 ±43.98 | 85.75 % ±5.03 % | 355.95 ±65.61 | 86.44 % ±4.64 % |
| STREAM LENGTH | 343.86 ±29.33 | 86.17 % ±4.66 % | 311.32 ±43.91 | 85.72 % ±5.01 % | 393.39 ±81.15 | 86.37 % ±4.67 % |
| ELEVATION DIFFERENCE | 302.76 ±48.07 | 86.11 % ±4.69 % | 314.72 ±42.75 | 85.35 % ±5.28 % | 411.96 ±80.55 | 86.33 % ±4.71 % |
| AVERAGE SLOPE | 276.88 ±40.39 | 86.08 % ±4.67 % | **279.22** ±**41.44** | 85.44 % ±5.32 % | 364.96 ±79.10 | 86.26 % ±4.79 % |
| LEARNED | **169.93** ±**33.40** | 86.14 % ±4.87 % | 280.07 ±46.97 | **86.03 %** ±**4.80 %** | 323.54 ±83.12 | **86.48 %** ±**4.69 %** |

Table 3: Pearson correlation between learned and physical edge weights.

| | LEARNED EDGE WEIGHTS | | | | | |
|---|---|---|---|---|---|---|
| | DOWNSTREAM | | UPSTREAM | | BIDIRECTIONAL | |
| PHYSICAL EDGE WEIGHTS | RESGCN | GCNII | RESGCN | GCNII | RESGCN | GCNII |
| STREAM LENGTH | 0.221 ±0.098 | −0.23 ±0.012 | 0.042 ±0.008 | −0.14 ±0.006 | −0.002 ±0.016 | 0.054 ±0.034 |
| ELEVATION DIFFERENCE | 0.100 ±0.021 | −0.17 ±0.003 | −0.308 ±0.015 | 0.027 ±0.007 | −0.235 ±0.014 | −0.103 ±0.035 |
| AVERAGE SLOPE | 0.168 ±0.038 | −0.04 ±0.007 | −0.293 ±0.009 | 0.090 ±0.009 | −0.24 ±0.012 | −0.163 ±0.012 |

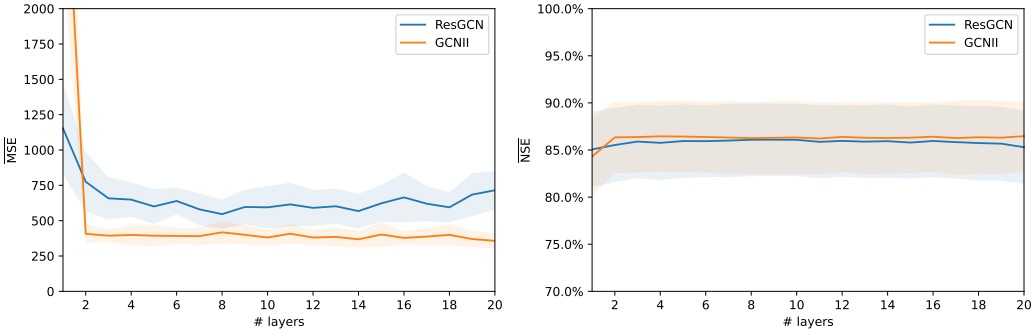

Figure 3: Model performance with varying depth, averaged over folds. Shaded areas correspond to $95\,\%$ confidence intervals across folds.

### 4.3 LEARNING THE WEIGHTS

The case of learned edge weights is of particular interest. They were initialized by drawing from the uniform distribution in $[0.9, 1.1]$ to arrange them neutrally around one while still introducing sufficient noise to break symmetry. Whenever learned weights get negative during training, we clip them to zero. The distribution of the learned weights (cp. Table A.3) is still centered around one with minima close to zero and maxima below ten.

To see if the learned weights exhibit any similarities with the physical weights, we calculate Pearson correlation coefficients for all topology combinations. Table 3 shows that none of the physical weight assignments correlate much with the learned weights. In multiple instances, the sign even flips when using a different model architecture. For instance, the largest positive correlation occurs with stream length for ResGCN, but in this same case GCNII achieves a negative correlation of the same magnitude. Hence, we conclude that none of the physical edge weights from the datasets are optimal context information for the predictor.

### 4.4 THE ROLE OF GNN DEPTH

The rationale for setting the number of layers to $N = 20$ was to allow information to propagate across the entire river network. However, since removing all edges from the graph does not deteriorate the performance (cp. Table 2), we can also consider shallower neural networks. In particular, we want to exclude the possibility that the considerable depth is causing the GCN to not outperform the baseline MLP due to more general issues with training very deep networks. In this case, a GCN with fewer layers could profit more from the graph structure despite not achieving global information propagation. Hence, we train ResGCN and GCNII with the default hyperparameters from Table 1 where we only vary the number of layers in steps of one from 1 to 20. The resulting average MSE and NSE scores are shown in Figure 3.

The experiment provides two insights. First, the inability of both GCN architectures to outperform the MLP baseline is consistent across network depths, so that we can rule out training issues. Second, the performance is independent of model depth, which means that the larger receptive field achieved by more layers does not help. Both corroborate the previous observations that GNNs fail to take advantage of the graph structure.

### 4.5 WORST GAUGE INVESTIGATION

The performance on gauge #80 of all trained models is considerably below the mean. For instance, the best overall performing model according to NSE (bidirectional-learned GCNII) achieves its worst NSE of only $24.78\,\%$ on this outlier gauge. To better understand the scenarios that are challenging for the model, we determine the top disjoint time horizons of 48 hours (24 hours for past and future) in terms of deviation of model prediction from the ground truth. The resulting plots in Figure 4 reveal that the outlier gauge is characterized by sudden spikes, which are inherently hard to forecast for any predictor. The gauge might be located behind a floodgate. As a result, the forecasting performance is mediocre, with the forecast often missing spikes.

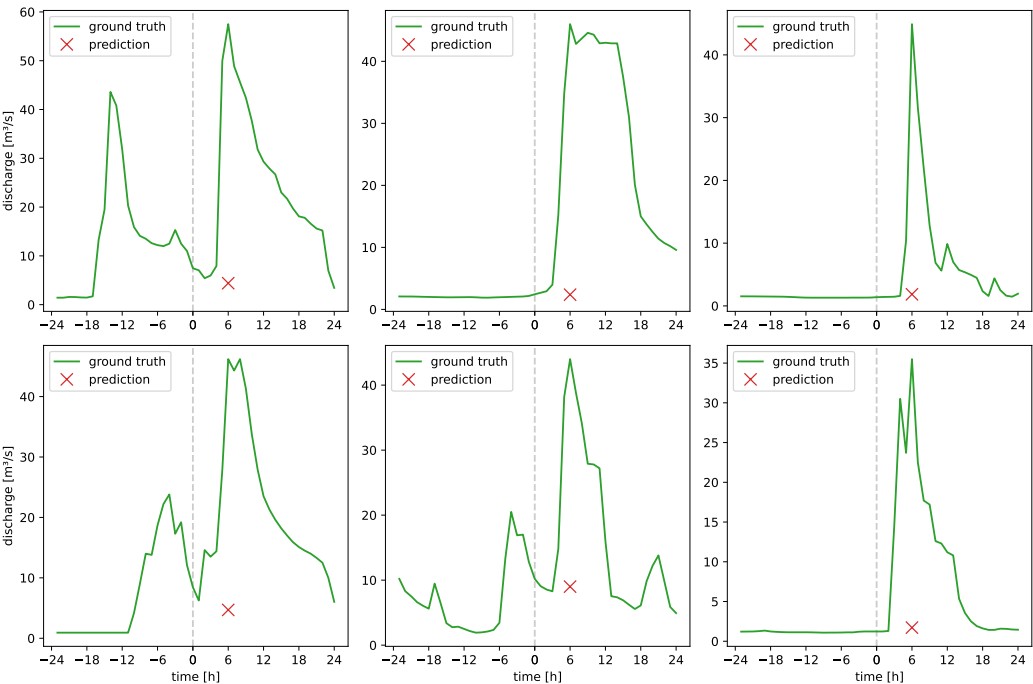

Figure 4: Worst predictions of bidirectional-learned GCNII on its overall worst gauge #80. Negative time indicates past, and positive time indicates future discharge.

## 5    CONCLUSION

In this work, we explored the applicability of GNNs to holistic flood forecasting in a river network graph. Based on the LamaH-CE dataset, we framed a supervised node regression task for predicting future discharge at all gauging stations in the graph given past observations. By modifying the adjacency matrix, we compared the impact of different adjacency definitions on the prediction performance. Our results reveal that the impact of river topology is negligible. The GNN performs equally well even when all edges are removed from the graph, which makes it act like an MLP. It does not benefit from weighted edges that resemble physical relationships between gauges. When the model is allowed to jointly learn the edge weights along with the other parameters, they correlate with neither constant weights nor any of the physical weightings given by the dataset. A depth study shows that the results are not caused by issues with training deep models but prove consistent throughout any number of layers. Investigations on a challenging outlier gauge show that the GNNs struggle to predict sudden discharge spikes.

On a high level, future work is encouraged to investigate under which conditions including graph topology in neural predictors actually helps, which is not clear a priori. While the key could lie in employing more specialized model architectures such as DGCN (Tong et al., 2020), MagNet (Zhang et al., 2021), and DAGNN (Thost & Chen, 2021) for the dataset at hand, there might be more fundamental limitations to the use of GNNs for large-scale regression problems. Moreover, for the application of flood forecasting, our results suggest that focusing on accurate spike prediction is more promising than incorporating river network topology information. To this end, there is a broader issue: we used a river network dataset from central Europe as discharge measurements are readily available there for long time periods. However, the regions most affected by floods are typically in low-income countries where data is scarce. More gauges need to be installed in those high-risk regions, and large-scale datasets collected to enable more relevant studies and save lives.

ACKNOWLEDGMENTS

(left out for blind review)

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

# A APPENDIX

## A.1 PREPROCESSING ALGORITHMS

---

**Algorithm A.1:** Inverse depth-first search

---

**Input:** DAG $\mathcal{G} = (\mathcal{V}, \mathcal{E})$, start node $v_0 \in \mathcal{V}$
**Output:** All direct and indirect predecessors of $v_0$ in $\mathcal{G}$

$\text{inverseDFS}(\mathcal{G}, v_0)$
1     $\mathcal{V}_{\text{in}} \leftarrow \{v \in \mathcal{V} \mid (v, v_0) \in \mathcal{E}\}$
2     **if** $\mathcal{V}_{\text{in}} = \emptyset$ **then**
3        **return** $\{v_0\}$
4     **else**
5        **return** $\{v_0\} \cup \bigcup_{v \in \mathcal{V}_{\text{in}}} \text{invDFS}(v)$

---

---

**Algorithm A.2:** Rewire-removal of a node

---

**Input:** antitransitive weighted DAG $\mathcal{G} = (\mathcal{V}, \mathcal{E}, w)$, moribund node $v_{\text{RIP}} \in \mathcal{V}$
**Output:** $\mathcal{G}$ without $v_{\text{RIP}}$ where its predecessors and successors are rewired

$\text{rewireRemove}(\mathcal{G}, v_{\text{RIP}})$
1     $\mathcal{V}_{\text{in}} \leftarrow \{v \in \mathcal{V} \mid (v, v_{\text{RIP}}) \in \mathcal{E}\}$
2     $\mathcal{V}_{\text{out}} \leftarrow \{v \in \mathcal{V} \mid (v_{\text{RIP}}, v) \in \mathcal{E}\}$
3     $\mathcal{V} \leftarrow \mathcal{V} \setminus \{v_{\text{RIP}}\}$
4     $\mathcal{E} \leftarrow \mathcal{E} \setminus (\mathcal{V}_{\text{in}} \times \{v_{\text{RIP}}\}) \setminus (\{v_{\text{RIP}}\} \times \mathcal{V}_{\text{out}}) \cup (\mathcal{V}_{\text{in}} \times \mathcal{V}_{\text{out}})$
5     **for** $(v_{\text{in}}, v_{\text{out}}) \in \mathcal{V}_{\text{in}} \times \mathcal{V}_{\text{out}}$ **do**
6        $w(v_{\text{in}}, v_{\text{out}}) \leftarrow w(v_{\text{in}}, v_{\text{RIP}}) + w(v_{\text{RIP}}, v_{\text{out}})$

---

## A.2 RESULTS FOR VANILLA GCN

Table A.2: Forecasting performance of a vanilla GCN on different river network topologies, given as mean and standard deviation of each metric across folds. A 20-layer MLP baseline achieves an NSE of $85.61\% \pm 4.90\%$. Note that results for the isolated adjacency type are not affected by the choice of edge orientation due to the absence of edges in this case.

| ADJACENCY TYPE | DOWNSTREAM | | UPSTREAM | | BIDIRECTIONAL | |
|---|---|---|---|---|---|---|
| | $\overline{\text{MSE}} \downarrow$ | $\overline{\text{NSE}} \uparrow$ | $\overline{\text{MSE}} \downarrow$ | $\overline{\text{NSE}} \uparrow$ | $\overline{\text{MSE}} \downarrow$ | $\overline{\text{NSE}} \uparrow$ |
| ISOLATED | 354.45 ±71.39 | 85.56% ±4.93% | 354.45 ±71.39 | 85.56% ±4.93% | 354.45 ±71.39 | 85.56% ±4.93% |
| BINARY | 4871.21 ±3464.82 | 28.79% ±21.49% | 5444.81 ±1363.50 | 33.98% ±17.36% | 4715.03 ±1359.65 | 69.80% ±8.08% |
| STREAM LENGTH | 3184.44 ±752.60 | 33.87% ±18.94% | 5041.33 ±1397.70 | 47.14% ±17.37% | 3778.37 ±575.68 | 76.74% ±5.75% |
| ELEVATION DIFFERENCE | 5316.44 ±3411.55 | 28.74% ±21.42% | 5577.10 ±1259.32 | 35.71% ±18.25% | 3132.41 ±799.16 | 78.17% ±5.87% |
| AVERAGE SLOPE | 9436.05 ±4272.67 | 10.32% ±30.85% | 5060.77 ±1535.16 | 34.76% ±19.29% | 4257.56 ±1619.34 | 72.59% ±8.65% |
| LEARNED | 1067.59 ±325.14 | 34.99% ±18.23% | 4750.12 ±1299.36 | 37.87% ±19.49% | 1868.82 ±533.35 | 75.48% ±7.22% |

## A.3 LEARNED EDGE WEIGHTS STATISTICS

Table A.3: Key statistics of the learned edge weights, accumulated across folds.

| | LEARNED EDGE WEIGHTS | | | | | |
| | DOWNSTREAM | | UPSTREAM | | BIDIRECTIONAL | |
| STATISTIC | RESGCN | GCNII | RESGCN | GCNII | RESGCN | GCNII |
| --- | --- | --- | --- | --- | --- | --- |
| MEAN | 0.989 ±0.013 | 0.768 ±0.002 | 0.666 ±0.011 | 0.793 ±0.008 | 0.917 ±0.006 | 0.955 ±0.008 |
| STD | 0.511 ±0.212 | 0.665 ±0.025 | 0.537 ±0.006 | 0.825 ±0.022 | 0.635 ±0.036 | 0.630 ±0.026 |
| MIN | 0.109 ±0.268 | 0.000 ±0.000 | 0.000 ±0.000 | 0.000 ±0.000 | 0.000 ±0.000 | 0.000 ±0.000 |
| 25% | 0.624 ±0.160 | 0.279 ±0.028 | 0.201 ±0.022 | 0.227 ±0.022 | 0.451 ±0.032 | 0.473 ±0.021 |
| MEDIAN | 1.042 ±0.019 | 0.599 ±0.021 | 0.588 ±0.026 | 0.570 ±0.015 | 0.851 ±0.024 | 0.919 ±0.017 |
| 75 % | 1.365 ±0.151 | 1.172 ±0.031 | 1.049 ±0.027 | 1.134 ±0.037 | 1.298 ±0.036 | 1.306 ±0.027 |
| MAX % | 3.257 ±0.983 | 5.463 ±0.895 | 2.217 ±0.052 | 6.772 ±0.489 | 3.197 ±0.256 | 3.515 ±0.286 |

