# OpenReview forum: "Exploiting River Network Topology for Flood Forecasting with Graph Neural Networks"
_ICLR.cc/2024/Conference — Submitted to ICLR 2024_

### Official Review · Reviewer_FSEP · 2023-10-31

**Soundness:** 2 fair
**Presentation:** 3 good
**Contribution:** 2 fair
**Rating:** 3
**Confidence:** 4

**Summary:**

The authors study the use of graph neural networks in flood forecasting. They find that with their models the use of river topology does not add significant value to the prediction of flood events.

**Strengths:**

- Flood prediction is an important problem under the climate change, and can lead to enormous social good via mitigation of the loss of human lives and economic damage.

- The current work applies some of the most advanced time-series prediction method like LSTM, combined with using GCN to take into account the network topology to solve this problem.

**Weaknesses:**

- The key issue with the current paper is the message is mostly a negative result, that incorporating the network structure does not help with the forecast of flood. Although there are some experimental support to this, it is difficult to draw this conclusion because the authors have only tried a limited set of models. As the claim is counter-intuitive, more analysis, especially analysis of the raw data, is required for supporting the claim. Are there correlations between the water level in two gauge stations upstream and downstream? And are there time lags between between a spike in an upstream gauge station and a downstream gauge station? If so, the spike in the upstream gauge station should be useful for predicting the spike in the downstream gauge station. Why is this not reflected in the experiment results?

- I have reservations about the way the problem is modeled. Since we are interested in predicting flood, which is a rare event, fitting the average MSE loss to the time series data might not be the best approach. Assuming the floods are rare large spikes in the data, a conservative model will do best by trying NOT to predict a large spike, as getting the timing of the spike wrong can incur a huge MSE loss. It could, for example, be modeled alternatively as a time series prediction problem, where we try to predict if there is a flood event within the next 6 hours given the water level in the past 24 hours. There could be multiple ways to model this but MSE on the water level does not seem the right fit for modeling rare spike events.

**Questions:**

- How long does it take on average for the flow from one upstream gauge station to reach a neighboring downstream gauge station? I think these values should be taken into account for the history window size (24 hours currently) and the prediction horizon (6 hours currently).

- What is the sample size for each gauge station? Or more importantly, what is the average number of flood events that each gauge station experience in the data? If the sample size is small, it could be beneficial to just pool all the data and train one model for the time series prediction of rare events, than to spread the rare event samples across multiple stations and try to model their correlations. Could that be a reason why the incorporation of network structure is not helping here?

- The authors claim that the methods perform similarly. From Table 2 it seems to be true for NSE, but MSE has a lot more variations across the different methods. Why is this the case?

---

> ### Author Response · Authors · 2023-11-22
> **Reply to Reviewer FSEP (1/2)**
>
> We are grateful the reviewer took the time to express detailed concerns and questions. It is encouraging that they find our research a relevant and state-of-the-art contribution. We hope to alleviate the reviewer’s reservations regarding modeling in the following.
>
> ---
>
> > I have reservations about the way the problem is modeled. Since we are interested in predicting flood, which is a rare event, fitting the average MSE loss to the time series data might not be the best approach. Assuming the floods are rare large spikes in the data, a conservative model will do best by trying NOT to predict a large spike, as getting the timing of the spike wrong can incur a huge MSE loss. It could, for example, be modeled alternatively as a time series prediction problem, where we try to predict if there is a flood event within the next 6 hours given the water level in the past 24 hours. There could be multiple ways to model this but MSE on the water level does not seem the right fit for modeling rare spike events.
>
> We agree that a rare event detection setup would be strictly preferable. However, such an approach requires ground-truth annotations of all flooding events, which the dataset unfortunately does not provide. Hence, the only option with the given data is regression of river discharge. On the other hand, the datasets that do have flood annotations are lacking the graph structure. We apologize for not having made this clear in the draft and updated Section 3.1 accordingly.
>
> ---
>
> > The key issue with the current paper is the message is mostly a negative result, that incorporating the network structure does not help with the forecast of flood.
>
> We believe that especially unintuitive negative results should be published to highlight the finding to the community. This may motivate other researchers to investigate the issue from their different viewpoints and complete the picture.
>
>
> ---
>
> > Although there are some experimental support to this, it is difficult to draw this conclusion because the authors have only tried a limited set of models.
>
> The set of models we employed is actually quite large: 3 major architectures times 18 topology combinations and each cross-validated on 6 different folds of a big dataset, which amounts to 324 models. This does not include the additional 240 models for the depth experiment. While we agree there are more architectures and choices that can be explored, we had to restrict our search space to retain computational feasibility. This issue would be mitigated by publishing our results so the community can address the problem with its accumulated computational resources.

---

> ### Author Response · Authors · 2023-11-22
> **Reply to Reviewer FSEP (2/2)**
>
> ---
>
> > Are there correlations between the water level in two gauge stations upstream and downstream?
>
> Physically, they are clearly correlated since they share the same water stream. Statistically, we also find linear cross-correlations, but the corresponding shift naturally varies.
>
> ---
>
> > How long does it take on average for the flow from one upstream gauge station to reach a neighboring downstream gauge station? I think these values should be taken into account for the history window size (24 hours currently) and the prediction horizon (6 hours currently).
>
> > And are there time lags between between a spike in an upstream gauge station and a downstream gauge station?
>
> We argue that the time lags are not relevant for prediction since the flow fluctuation and spikes will ultimately be reflected in the downstream gauge station's own discharge history.
>
> ---
>
> > What is the sample size for each gauge station? Or more importantly, what is the average number of flood events that each gauge station experience in the data? If the sample size is small, it could be beneficial to just pool all the data and train one model for the time series prediction of rare events, than to spread the rare event samples across multiple stations and try to model their correlations. Could that be a reason why the incorporation of network structure is not helping here?
>
> Pooling across all gauges, as suggested by the reviewer, would contract the river network graph into a singular point, thus making any topology considerations meaningless. There is a body of research on improving discharge prediction by training on pooled data from multiple gauges, but that caters to an entirely different research question than the one we are addressing.
>
> ---
>
> > The authors claim that the methods perform similarly. From Table 2 it seems to be true for NSE, but MSE has a lot more variations across the different methods. Why is this the case?
>
> Discharge scale across gauges varies massively. The MSE we show in Table 2 is not scale-normalized, while NSE is (cp. Section 3.2, Metrics). Hence, MSE is highly sensitive to even minor errors on gauges in broad waterbeds. However, we want the model to do equally well on all gauges of the river network. For this reason (among others), hydrology papers typically look at scale-normalized metrics such as NSE. We only included the non-scale-normalized MSE for completeness. Note that during training, the data is normalized, and thus, the optimization objective an implicitly scale-normalized MSE. We updated the caption of Table 2 to prevent any potential confusion in this regard.

---

### Official Review · Reviewer_T47T · 2023-11-04

**Soundness:** 2 fair
**Presentation:** 4 excellent
**Contribution:** 3 good
**Rating:** 5
**Confidence:** 3

**Summary:**

The paper described an application of graph neural nets for making using of river network topology, for the task of forecasting river discharge. The paper reported under different settings, incorporating such topology information does not improve the prediction accuracy and concludes that topology information doesn’t add additional prediction capacity for this task.

**Strengths:**

1. The paper is well-written and include in-depth details about the experiments, models and data. Together with publicly available code, the result seems to be highly reproducible.
2. The paper discuss the learnings from introducing topology information in a credible fashion, which contributes more to the community how and when such information is not useful.

**Weaknesses:**

My biggest concern with the paper is that, for such time series forecasting tasks, the paper uses cross validation instead of backtesting for validating model performance. Specifically, the authors splitter 18 years of data into 6 folds of 3 years of data and conducted cross-validation. I found it hard to be convinced of any conclusion derived from such setup. For example, in one of the 6 runs where training data happen to include data in 2014 and the prediction data happens to include data in 2013 - in that case, a model trained with future information is used to predict the past - unless there is no auto-correlation at all, otherwise, such future leakage will undermine the conclusion made from such numerical results. With such concern, it may not necessary be that topology information does not contribute to forecasting future river discharge, but rather, since the model has already known about the past and the future about the river discharge, such topology information may not add any information any more. In practical settings where only historical information is known, having such topology information could still yield accuracy gain - well this is just a hypothesis assuming the experiment is properly set up such that no future information is leaked.

**Questions:**

My suggestion is to re-run the experiment with backtest settings instead of cross-validation settings.

---

> ### Author Response · Authors · 2023-11-22
> **Reply to Reviewer T47T**
>
> We thank the reviewer for taking the time to navigate our rather involved setup. It is encouraging that they view our contribution, its presentation, and reproducibility as valuable to the community. The reviewer's only concern is with regard to our validation process, which we would like to untangle in the following.
>
> ---
>
> > For example, in one of the 6 runs where training data happen to include data in 2014 and the prediction data happens to include data in 2013 - in that case, a model trained with future information is used to predict the past - unless there is no auto-correlation at all, otherwise, such future leakage will undermine the conclusion made from such numerical results.
>
> Indeed, with the vanilla cross-validation setup, some test years lie before the training years. In this vein, the model is also assessed as to its performance on past data it has not seen. This is a valid desideratum and has nothing to do with any "future leakage" since we are not trying to build a production model but are interested in the general effect of topology information on predictive capabilities from a data analysis standpoint. The only possible source of actual future leakage would be in the construction of supervised input-output samples; however, our sampling strategy enforces the temporal relationship of "input < output" (cp. Section 3.2, Features & Targets).
>
> ---
>
> > With such concern, it may not necessary be that topology information does not contribute to forecasting future river discharge, but rather, since the model has already known about the past and the future about the river discharge, such topology information may not add any information any more.
>
> While it is true that in this case, the model has seen data from past and future years, it has not seen any data from the current year it is tested on, so the graph structure can still add information at prediction time, as it makes a portion of the current year's data (the past 24 hours) from neighboring gauges available.
>
> ---
>
> > My suggestion is to re-run the experiment with backtest settings instead of cross-validation settings.
>
> We would like to note that backtesting is, in fact, the *opposite* of what the reviewer describes. By definition, backtesting refers to validating the model on only *past* inputs to verify that it would have also predicted past events accurately. Its wide use in financial time series analysis confirms that testing on the past is not an issue per se.

---

### Official Review · Reviewer_9HHN · 2023-11-21

**Soundness:** 2 fair
**Presentation:** 3 good
**Contribution:** 4 excellent
**Rating:** 5
**Confidence:** 4

**Summary:**

The authors conducted an experiment to determine if topology information (i.e. slope, elevation change, and distance) could be used to improve flood forecasting results for graph neural networks. The authors concluded that the models do not benefit from inclusion of this information.

**Strengths:**

The field of flood forecasting still relies heavily on physics-based models and there is excellent potential to implement machine-learning approaches to improve existing practices.

The authors provide a detailed description of the methodology that was implemented, and the results should be reproducible by other researchers.

Relevant metrics to the field of flood forecasting including the Nash-Sutcliffe Efficiency.

**Weaknesses:**

Aside from Figure 4, which is described as the worst case scenario, no graphical results are shown to demonstrate the behaviour of the model.

If the model is intended to be used for flood forecasting rather than general daily flow forecasting, the training and testing process used in this study may not be appropriate. Instead of evaluating an entire period of record, flood forecasting models typically identify individual events within the historical flow record using an approach such as peak over threshold.

Given the negative results, additional attempts should be made to identify and demonstrate how/why the model is failing. Graphical analysis of results for a few sites would help to evaluate this.

**Questions:**

It seems that the choices of 6 hours and 24 hours for the lead time and lookback windows, respectively, are arbitrary. The flood wave celerity between stations under typical flood peaks could be computed from the historical flow records to make an informed decision on how to select this window, ensuring that peaks have enough time to travel between stations. Do the selected values allow for appropriate travel time for most stations?

What did the results look like on the best-performing stations? Were flood peaks predicted with errors in timing or magnitude, or did the model fail to predict them at all?

Is it reasonable to perform element-wise normalization for a graph where the value of nodes relative to adjacent nodes is important? Additionally, have you considered that these absolute values (i.e. flows) may affect flood wave celerity (i.e. changes in discharge under Manning’s equation)?

---

> ### Author Response · Authors · 2023-11-22
> **Reply to Reviewer 9HHN**
>
> We appreciate the reviewer’s positive feedback on the relevance of our work and their detailed questions coming from a physics/hydrology angle.
>
> ---
>
> > If the model is intended to be used for flood forecasting rather than general daily flow forecasting, the training and testing process used in this study may not be appropriate. Instead of evaluating an entire period of record, flood forecasting models typically identify individual events within the historical flow record using an approach such as peak over threshold.
>
> The problem with approaches like peak over threshold is that they assume one can tell a flood event solely from the discharge value. However, the presence of flooding does not necessarily depend only on discharge but many other factors like soil moisture and the height of river dikes which, to make matters worse, do change over time. Hence, it is totally unclear how to set the threshold hyperparameter appropriately. A proper event-focused approach would only work if we had expert flood annotations as part of the dataset, which is not the case (see also reply to Reviewer FSEP).
>
> ---
>
> > Aside from Figure 4, which is described as the worst case scenario, no graphical results are shown to demonstrate the behaviour of the model.
>
> > Given the negative results, additional attempts should be made to identify and demonstrate how/why the model is failing. Graphical analysis of results for a few sites would help to evaluate this.
>
> It seems questionable how a qualitative inspection of forecast plots would help in identifying the cause of the topology-informed models failing to outperform the baseline. Since we found the models for all topology combinations to perform very similarly, those plots consequently look very similar, and there is hardly anything to infer from them. Thus, we only included plots for the worst-case gauge that highlight the issues all of the model combinations struggle with.
>
> ---
>
> > It seems that the choices of 6 hours and 24 hours for the lead time and lookback windows, respectively, are arbitrary. The flood wave celerity between stations under typical flood peaks could be computed from the historical flow records to make an informed decision on how to select this window, ensuring that peaks have enough time to travel between stations. Do the selected values allow for appropriate travel time for most stations?
>
> They are rather arbitrary but chosen high enough to allow for appropriate information propagation. Flood wave celerity could only be computed from the records if we had flood annotations, which is not the case (see also reply to Reviewer FSEP).
>
> ---
>
> > What did the results look like on the best-performing stations? Were flood peaks predicted with errors in timing or magnitude, or did the model fail to predict them at all?
>
> Since the dataset does not provide flood annotations, we cannot know which peaks in the discharge time series are associated with floods and which are not. In favor of conducting responsible research, we refrained from making any blunt assumptions along the lines of “discharge above x implies flood”. The consequence is that we cannot look at individual flooding events but only at the regression performance.
>
> ---
>
> > Is it reasonable to perform element-wise normalization for a graph where the value of nodes relative to adjacent nodes is important? Additionally, have you considered that these absolute values (i.e. flows) may affect flood wave celerity (i.e. changes in discharge under Manning’s equation)?
>
> What matters is the fluctuations of the neighbor’s features (discharge and meteorological indicators) over time rather than the absolute value. If we only applied some global normalization to all gauges, we would run into severe numerical issues due to the vastly differing feature scale across gauges.

---

### Comment · Area_Chair_UTtT · 2023-11-21
**New review added**

Just a heads up to the authors that a new review has been added. We apologize for the late notice here.

---

### Author Response · Authors · 2023-11-22
**Summary Reply**

All reviewers appreciate the paper’s high-quality presentation (scores 4/3/3), specifically the degree of reproducibility. Furthermore, they acknowledge that our contribution adds true value (scores 4/3/2). There were only reservations with regard to soundness (scores 2/2/2), which we hope to have alleviated in our individual responses below. In this general comment, we only highlight two misconceptions that seem to be shared between multiple reviews.

The first misconception is that the dataset would contain expert flood annotations which specify the timings of all flood events. However, this is not the case, and thus, many points that the reviewers raise concerning the possibility of event-based modeling and matching flood peaks of neighboring gauges become void.

The second misconception is that the model input would only be river discharge for the past 24 hours. However, we also feed meteorological indicators for those hours, particularly precipitation, topsoil moisture, air temperature, and surface pressure. We apologize if this was unclear and updated Section 3.2 to explain it more. This and all other modifications to the paper are easily recognizable by their blue text color.

---

### Meta-Review · Area_Chair_UTtT · 2023-12-09

**Metareview:**

This paper attempts to predict river discharge using GNNs to incorporate the river topology. The authors find that GNNs are not helpful in this case across a range of model architectures. While this application is useful and a negative result here would certainly be of use and interest to the community, the reviewers believe that the conclusions are not adequately supported in the present draft, and I must therefore recommend rejection.

**Justification For Why Not Higher Score:**

The reviewers concur this paper should not be accepted.

**Justification For Why Not Lower Score:**

n/a

---

### Decision · Program_Chairs · 2024-01-16

Reject